# Genome and Transcriptome Identification of a Rice Germplasm with High Cadmium Uptake and Translocation

**DOI:** 10.3390/plants12061226

**Published:** 2023-03-08

**Authors:** Jin-Song Luo, Bao Guo, Yiqi He, Chun-Zhu Chen, Yong Yang, Zhenhua Zhang

**Affiliations:** 1College of Resources, Hunan Agricultural University, Changsha 410128, China; 2Hunan Provincial Key Laboratory of Farmland Pollution Control and Agricultural Resources Use, Hunan Provincial Key Laboratory of Nutrition in Common University, National Engineering Laboratory on Soil and Fertilizer Resources Efficient Utilization, Changsha 410128, China; 33D Medicines, Block A, Building 2, No.158 Xinjunhuan Road, Pujiang Town, Minhang District, Shanghai 201210, China

**Keywords:** cadmium accumulation, cell wall, pectin methylesterase, rice, phytoremediation, multi-omics

## Abstract

The safe production of food on Cd-polluted land is an urgent problem to be solved in South China. Phytoremediation or cultivation of rice varieties with low Cd are the main strategies to solve this problem. Therefore, it is very important to clarify the regulatory mechanism of Cd accumulation in rice. Here, we identified a rice variety with an unknown genetic background, YSD, with high Cd accumulation in its roots and shoots. The Cd content in the grains and stalks were 4.1 and 2.8 times that of a commonly used japonica rice variety, ZH11, respectively. The Cd accumulation in the shoots and roots of YSD at the seedling stage was higher than that of ZH11, depending on sampling time, and the long-distance transport of Cd in the xylem sap was high. Subcellular component analysis showed that the shoots, the cell wall, organelles, and soluble fractions of YSD, showed higher Cd accumulation than ZH11, while in the roots, only the cell wall pectin showed higher Cd accumulation. Genome-wide resequencing revealed mutations in 22 genes involved in cell wall modification, synthesis, and metabolic pathways. Transcriptome analysis in Cd-treated plants showed that the expression of pectin methylesterase genes was up-regulated and the expression of pectin methylesterase inhibitor genes was down-regulated in YSD roots, but there were no significant changes in the genes related to Cd uptake, translocation, or vacuole sequestration. The yield and tiller number per plant did not differ significantly between YSD and ZH11, but the dry weight and plant height of YSD were significantly higher than that of ZH11. YSD provides an excellent germplasm for the exploration of Cd accumulation genes, and the cell wall modification genes with sequence- and expression-level variations provide potential targets for phytoremediation.

## 1. Introduction

Owing to the rapid development of industrialization in both urban and rural areas of China, the Cd pollution levels in some farmlands exceed China’s national standard of 0.2 mg kg^−1^ (GB2762-2005), leading to Cd rice production, which poses a potential threat to human health. Cd in the human body can affect human reproduction and cause Itai-itai Disease [1]. Breeding low-Cd crop varieties and the phytoremediation of Cd-contaminated farmland are two important coping strategies [2]. Therefore, it is of great practical significance to elucidate the accumulation mechanism of Cd in rice.

Phytoremediation, in which metal pollutants of soil, such as Cd, can be absorbed by natural or artificially bred special plants through their roots to remove Cd or reduce their bioavailability in soil [3,4], which mainly includes plant extraction, plant stabilization, plant filtration, and plant stimulation [5,6]. Such methods are economically feasible, environmentally friendly, and highly applicable.

At present, Cd uptake and transport in rice is mainly thought to be conducted by the transporters of some essential metal elements. For example, the NRAMP (natural resistance-associated macrophage protein) family, NRAMP5, which is responsible for Mn uptake [7], NRAMP1, which is responsible for Fe and Mn uptake [8,9], and Zip5/9, which is responsible for Zn uptake, all contribute to the uptake of Cd [10,11]. Under the action of the P1B-type ATPases, known as heavy metal ATPases (HMAs), HMA3, some of the Cd taken up by the roots is stored in vacuoles [12,13]. Under the action of HMA2, some of it is transported to the shoots via long-distance transport in the xylem sap [14,15]. The Cd in the shoots is distributed from the nodes to the grains under the action of a low-affinity cation transporter, LCT1 [16]. Recently, OsCD1, belonging to the major facilitator superfamily, was found to be involved in root Cd uptake and contribute to grain accumulation in rice. Natural variation in OsCd1 with a missense mutation, Val449Asp, is responsible for the divergence of rice grain Cd accumulation between indica and japonica [17]. The rice defencin gene, CAL1, can chelate cytoplasmic Cd and efflux it into the xylem sap for long-distance transport to the leaves for storage [18].

The cell wall plays an important role in the detoxification and hyperaccumulation of Cd in *Sedum plumbizincicola* [19]. It has been reported that the cell wall can bind Cd, preventing Cd from entering the cytoplasm and reducing the toxicity of Cd in Arabidopsis [20]. When pectin in the cell wall is demethylated, it exposes the Cd ions to more carboxyl binding, playing a major role in Cd detoxification in *Brassica napus* [21]. The application of boron can decrease Cd content in plant tissues and improve the antioxidative system whilst increasing the Cd content in the cell wall, thus alleviating Cd and oxidation stress in rice [22]. Salicylic acid remarkably decreased Cd concentrations in roots and shoots of rice seedlings, and increased the distribution ratio of Cd in the root cell wall fraction through regulating root cell wall composition via nitric oxide signaling [23]. This study aimed to identify the physiological and molecular mechanisms of a rice germplasm YSD with high Cd uptake and translocation, which potentially provide valuable materials and genetic resources for phytoremediation.

## 2. Results

### 2.1. YSD Accumulated More Cd in the Straw and Grains Than ZH11

At the harvest stage of the rice, the Cd content in the rice straw and grains of YSD was significantly higher than that of ZH11, and the Cd accumulation in the rice straw and grains of YSD was 2.8 times and 4.1 times that of ZH11, respectively (Figure 1A,B). However, there were no significant differences in other elements in rice straw and grains, except for Ca, which was 54% lower in the YSD straw than in the ZH11 straw (Figure 1C,D).

### 2.2. Time-Dependent Higher Cd Accumulation in the Shoots and Roots of YSD

To further determine the phenotype of the high Cd accumulation in YSD, we treated rice with 10 μM Cd for 1, 3, and 5 days. The results showed that the Cd accumulation in the shoots and roots of YSD was significantly and time-dependently higher than that of ZH11. The Cd accumulation of YSD was 1.4, 1.6, and 1.7 times that of ZH11 in the shoots after Cd treatment for 1, 3, and 5 days, respectively (Figure 2A); it was 1.2, 1.2, and 1.5 times that of ZH11 in the roots (Figure 2B). We detected the Cd content in xylem sap treated with 10 μM Cd for 1 day and found that the concentration of Cd in the xylem sap of YSD was significantly higher (2.1 times higher) than that of ZH11 (Figure 2C). In addition, the Cd content at the junction of the root and stem of YSD was 25% higher than that of ZH11 (Figure 2D).

### 2.3. Cd Content in Various Subcellular and Cell Wall Components of YSD and ZH11

The Cd content in the shoot cell wall, organelle, and soluble fractions of YSD was 2.3, 1.7, and 1.5 times higher, respectively, than that of ZH11 (Figure 3A–C), representing a significant difference. The Cd content in the YSD roots was 59.8% higher than that in the ZH11 roots for the cell wall fraction, but there were no significant differences in the Cd content in the organelle and soluble fractions (Figure 3A–C). When we detected the Cd content in different components of the cell wall, the results showed that the Cd content in the shoot cellulose, hemicellulose, and pectin of YSD was 3.4, 1.9, and 3.6 times higher, respectively, than that of ZH11, representing a significant difference (Figure 3D–F). The Cd content in the pectin in the YSD roots was 223% higher than that in ZH11 roots, but there were no significant differences in the Cd content in the cellulose or hemicellulose in the roots (Figure 3D–F). These results indicate that the cell wall, organelles, and soluble fractions of YSD shoots all possess characteristics of high Cd accumulation, while the high Cd accumulation in the roots may be mainly determined by the cell wall pectin.

### 2.4. Genome-Level Variation of YSD Compared with ZH11

To study the effect of YSD genome-level variation on Cd accumulation, genome resequencing was conducted. The results showed that, compared with ZH11, there were 968,795 SNP sites and 185,084 InDel sites in the whole YSD genome (Figure 4A). These SNPs resulted in non-synonymous mutations in the coding regions of 5006 genes, the premature termination of 363 genes, the termination codon loss of 112 genes, and synonymous mutations in the coding regions of 3844 genes (Figure 4B). The InDel loci resulted in frameshift mutations in the coding regions of 1498 genes, non-frameshift mutations in 1258 genes, the premature termination of 84 genes, and the loss of termination codons in 12 genes (Figure 4C). KEGG pathway analysis of these affected genes revealed that 156 are carbohydrate-metabolism-related genes, 92 are lipid-metabolism-related genes, and 53 are membrane transport- and catalysis-related genes (Figure 4D). GO enrichment found 22 mutated genes related to cell wall synthesis and modification (Table 1).

### 2.5. Differentially Expressed Genes between YSD and ZH11 Roots

To reveal the differentially expressed genes in response to Cd in YSD roots, transcriptome sequencing was performed on ZH11 and YSD roots. Principal component analysis showed that the first principal component could explain 84.83% and the second 8.7% of the gene expression differences in the roots of the two cultivars (Figure 5A). There were 602 up-regulated and 297 down-regulated genes in the ZH11 roots, and 584 up-regulated and 297 down-regulated genes in the YSD roots when exposed to Cd treatment (Figure 5B). Compared to ZH11, 972 genes in YSD were up-regulated and 1073 down-regulated under Cd treatment (Figure 5B). Under control treatment conditions, 965 YSD genes were up-regulated and 1207 down-regulated (Figure 5C). A Venn diagram showed that 354 genes responded to the change in Cd in both YSD and ZH11 (Figure 5C), while 545 genes responded only in ZH11, and 527 genes responded only in YSD. Under both, Cd treatment and control conditions, 1419 differentially changed genes were consistent in YSD and ZH11. There were 778 genes that were differentially expressed only under control conditions, and 625 genes that responded only to Cd treatment. Finally, 24 genes differed in all four comparisons (Figure 5C). Figure 5D shows a heat map of differentially expressed genes under Cd treatment. We conducted GO enrichment analysis on the differentially expressed genes, and the results showed that the differentially expressed genes in the ZH11 roots were mainly enriched in the process of nicotinamide synthesis, ethylene signal transduction, and ammonium transmembrane transport. The differentially expressed genes were mainly located in the extracellular apoplast (Figure 5E). The differentially expressed genes in the YSD roots were mainly enriched in GSH metabolism, iron and superoxide balance, and the differentially expressed genes were mainly located in the extracellular region of the cell wall (Figure 5F).

### 2.6. Nicotinamide Promotes Cd Translocation in Rice

According to the GO enrichment results of ZH11 in response to Cd, we found that the expression of the nicotinamide synthesis genes, NAS1 and NAS2, in ZH11 under Cd treatment was 3.5 and 3.6 times higher, respectively, than that in YSD (Appendix A). The nicotinamide content in the ZH11 roots and shoots was 37% and 21% higher, respectively, than that in YSD. In ZH11 (Appendix A), the expression of YSL2, which is responsible for NA–Fe transport, was 592 times higher than that of YSD (Appendix A). Based on the Cd accumulation characteristics of ZH11 and YSD, as well as this result, we speculated that nicotinamide synthesis and YSL2-mediated metal chelate transport may be negatively correlated with Cd accumulation. We designed an in vitro experiment involving the addition of 0, 10, and 100 µM nicotinamide to a rice solution, and found that in vitro, 10 and 100 µM nicotinamide treatment increased the Cd accumulation in ZH11 shoots by 175% and 195% (Appendix A), and in YSD shoots by 165% and 175%, respectively (Appendix A). This is contrary to our speculation, indicating that the differential accumulation characteristics of YSD and ZH11 Cd may not be caused by the differences in nicotinamide synthesis.

According to the GO enrichment results of Cd in YSD, we found that GSH transferase gene expression in YSD under Cd treatment was 2–3 times higher than that in ZH11, and the GSH content in the roots and shoots was significantly lower (37% and 30%) than that in ZH11 (Appendix A). This result indicates that GSH metabolism may be involved in the different Cd accumulation and tolerance characteristics of YSD and ZH11.

### 2.7. No Significant Changes in Cd Uptake and Transport-Related Genes

In addition, we analyzed the expression of the currently reported genes responsible for Cd uptake, translocation, and vacuolar storage, and found that the genes responsible for Cd uptake (*NRMP1/NRAMP5*/*ZIP5*), long-distance transport (*HMA2*/*CAL1*), and vacuolar storage (*HMA3*) did not differ significantly between YSD and ZH11 (Appendix A), except for the significantly down-regulated expression of the Cd uptake gene *ZIP9* in YSD roots (Appendix A). These results indicate that the high Cd accumulation characteristics of YSD may not be caused by changes in the expression of these genes.

### 2.8. Expression of Pectin Modification and Metabolism-Related Genes

Previous studies found that differences in Cd accumulation between YSD and ZH11 roots were mainly determined by cell wall pectin. We analyzed the expression of genes related to pectin modification and catabolism. In YSD roots, the expression levels of PME genes (*PME22*/*PME35*/*PME67*/*PME32*) and pectin-synthesis-related genes (*Os05g0385116*/*GAUT3/BYM1*) were significantly higher than in ZH11 roots (Figure 6A), while those of PME-inhibitor genes (*PMEI8*/*PMEI28*) and pectin cleavage-related genes (*Os05g0213900/PAE5*) were significantly lower (Figure 6A). The PME activity was 51% higher in YSD than in ZH11 roots after Cd treatment (Figure 6B). These results reveal that pectin modification plays an important role in determining root Cd accumulation.

### 2.9. Comparison of Agronomic Characteristics between YSD and ZH11

We compared the agronomic traits of YSD and ZH11. The plant height of YSD was 175% higher than that of ZH11 (Figure 7A) and the straw dry weight of YSD was 3.15 times than that of ZH11 (Figure 7B). There were no significant differences in the tiller number or yield per plant between the two cultivars (Figure 7C,D).

## 3. Discussion

A wild rice variety, YSD, with high Cd accumulation at the seedling and harvest stages, was identified (Figure 1 and Figure 2). Agronomic trait analysis showed that the plant height and straw dry weight of YSD were significantly higher than that of ZH11 (Figure 7). Genome resequencing showed that the mutant genes were significantly enriched in genes related to carbon metabolism (Figure 4D), which might be the reason for the higher biomass of YSD.

The cell wall is the first defense barrier for plant cells. Many studies have shown that different components of the cell wall can chelate Cd and reduce the toxicity of Cd to cells [19,20,21]. Compared to ZH11, the cell wall components, soluble fraction, and organelles of YSD shoots showed obvious high Cd accumulation characteristics, while only the cell walls in the root showed high Cd accumulation (Figure 3). Genome-wide resequencing showed that 22 genes related to the synthesis and modification of YSD cell walls were mutated (Table 1). Pectin-modifying genes in the roots were significantly changed: the expressions of pectin methylesterase genes and pectin synthesis genes were up-regulated, while those of pectin-inhibiting genes and lytic-related genes were down-regulated (Figure 6). These results suggest that YSD cell wall plays an important role in high Cd accumulation, which may be caused by the enhancement of the Cd apoplast transport pathway.

The transcriptomic analysis of genes responsible for Cd uptake, long-distance transport, and vacuolar storage in YSD roots showed no significant changes compared to ZH11 (Appendix A), suggesting that the high accumulation of Cd in YSD may not be caused by these reported genes. Transcriptomics revealed that, when YSD responds to Cd stress, the GSH metabolic pathway is significantly enriched, and the GSH content in YSD plants is significantly reduced (Appendix A). GSH maintains the redox balance in YSD cells as a precursor of phytochelatin synthesis, and two proteins of the ABC family in Arabidopsis mediate the vacuole storage of the phytochelatin–Cd complex [24,25,26,27]. Currently, the gene that mediates PC–Cd transport in rice has not been identified; thus, it may also be mediated by the ABC family. Therefore, YSD provides valuable germplasm resources for the subsequent cloning of new Cd accumulating rice varieties.

Nicotinamide, as a ligand of some metal elements including Fe, Mn, and Cu, mediates the absorption and transport of NA–metal complexes, with the participation of YSL transporters [28]. Our study found that the expression level and content of nicotinamide synthesis genes in YSD were significantly lower than those in ZH11, and the expression level of YSL12 was significantly lower than that in ZH11 (Appendix A). These results suggest that the nicotinamide content and YSL12 expression in rice may be negatively correlated with Cd transport. In vitro nicotinamide treatment increased Cd accumulation in rice shoots (Appendix A), but had no effect on Cd uptake in the roots (Appendix A), suggesting that nicotinamide had different effects on Cd uptake and transport in vivo and in vitro.

## 4. Conclusions

In summary, we identified YSD, a rice variety with high Cd accumulation that provides excellent germplasm resources for the exploration of Cd accumulation-related genes, and the cell wall modification genes with sequence- and expression-level variations that provide potential targets for phytoremediation. Moreover, our findings suggest that the apoplast cell wall may play a key role in the accumulation of Cd.

## 5. Materials and Methods

### 5.1. Plant Materials, Analysis of Agronomic Characters and Growth Conditions

Zhonghua 11 (ZH11) is a commonly used japonica rice variety; whereas, YSD is a rice germplasm collected from Huaihua city with unclear genetic background. The rice was grown in paddy fields contaminated with 0.4 mg/kg Cd in Changsha, until the seeds were harvested. At the mature stage, 10 rice plants were randomly selected from YSD and ZH11 varieties, and the plant height, straw dry weight and yield per plant were measured, respectively.

The conditions for the hydroponic cultivation of the rice in the greenhouse were as follows: the seeds were soaked in water at 30 °C for 36 h in the dark, and the seeds with good germination were planted in 96-well bottomless plates. Rice seedlings were hydroponically cultivated, as previously described [18].

### 5.2. Plant Sampling and Elemental Determination

At 4 weeks, the hydroponically grown rice seedlings were exposed to the Cd treatments indicated in the following section before being sampled. The xylem sap collected from five plants was pooled into one replicate, and a total of three replicates were used for each line, as previously described [18]. The metal content was determined using inductively coupled plasma mass spectrometry (ICP-MS), as previously described [29].

In vitro niacinamide treatment was performed on 4-week-old hydroponic rice seedlings treated with 10 μM Cd, 10 μM Cd + 10 μM niacinamide and 10 μM Cd + 100 μM niacinamide for 3 days, respectively. The content of Cd in the shoots was determined by ICP-MS.

### 5.3. Extraction of Subcellular and Cell Wall Components and Determination of Cd Content

At 4 weeks, the rice plants were treated with 10 µM CdCl_2_ for 3 days. The shoots and roots of the plants were then collected. The subcellular components were extracted by differential centrifugation, as previously described [17]. The extracted cell wall and organelle fractions were dried in an oven, and after drying, the cell wall, organelle, and soluble fractions were digested in 70% HNO_3_ (*v*/*v*), before the Cd concentration was determined by ICP-MS.

Pectin, cellulose, and hemicellulose were extracted according to the methods, as previously described [21]. The Cd concentrations were determined by ICP-MS after dilution. Pectin methylesterase (PME) activity was determined, as previously described [20].

### 5.4. Whole-Genome Resequencing

Fresh leaves from 10-day-old plants were sampled for the isolation of genomic DNA. An Illumina HiSeq 4000 system (read length 350–500 bp, paired end) belonging to the OE Biotech company (Shanghai, China) was used to perform whole-genome resequencing to distinguish variations in the genomic DNA [30]. A total of 17G of data was generated, covering 40 times the rice genome. Genome-wide single-nucleotide polymorphisms (SNPs) and insertions/deletions (InDels) were identified and characterized between ‘YSD’ and ‘ZH11’ by the OE Biotech company (Shanghai, China).

### 5.5. RNA-SEQ

The roots of ZH11 and YSD were cultured for 28 days and sampled after 10 µM Cd treatment for 3 days. Three plant roots were sampled as replicates and quickly submerged in liquid nitrogen and sent to the OE Biotech company (Shanghai, China) for RNA sequencing analysis. Each replicated sample produced an average of 6G data. Three biological replicates were set. Heat maps of the differentially expressed genes were constructed using the MeV software (http://www.tm4.org/, 8 July 2021).

### 5.6. Determination of Glutathione and Nicotinamide Content

The glutathione (GSH) and nicotinamide content were determined by Suzhou Comin Biotechnology Co., Ltd. Specifically, the GSH in the roots and shoots was measured using a kit (GSH-1-W), with the o-phthalaldehyde fluorescence derivatization method. Nicotinamide was detected as described, with minor modification [31]. First, approximately 0.1 g of the sample was weighed out and 1 mL of mobile phase A was added. The mixture was then homogenized in an ice bath and an ice-bath ultrasound was conducted for 30 min. The sample was centrifuged at 8000× *g* for 10 min and the supernatant was collected, filtered using a needle filter, and subjected to high-performance liquid chromatography. A Rigol L-3000 system with a CST Daiso C18 (250 mm × 4.6 mm, 5 μm) column was used. Mobile phase A was 0.05 mol/L sodium acetate aqueous solution (acetic acid was used to adjust the pH to 4.5) and mobile phase B was methanol. The ratio of A to B was 90:10. The injection volume was 10 μL, the flow rate was 1 mL/min, the column temperature was 35 °C, the sample departure time was 30 min, and a UV detector wavelength of 261 nm was used for the method group.

### 5.7. Statistical Analysis and Data Availability

The data were analyzed using minimum differential multiple-range comparisons with the SPSS software, and each experiment was carried out with at least three biological replicates. *p* < 0.05 was considered to indicate a significant difference, and *p* < 0.01 was considered to indicate a highly significant difference. Charts were prepared with GraphPad Prism 8. The raw data for the whole-genome resequencing and mRNA transcriptome sequencing are available from the corresponding author on request.

## Figures and Tables

**Figure 1 plants-12-01226-f001:**
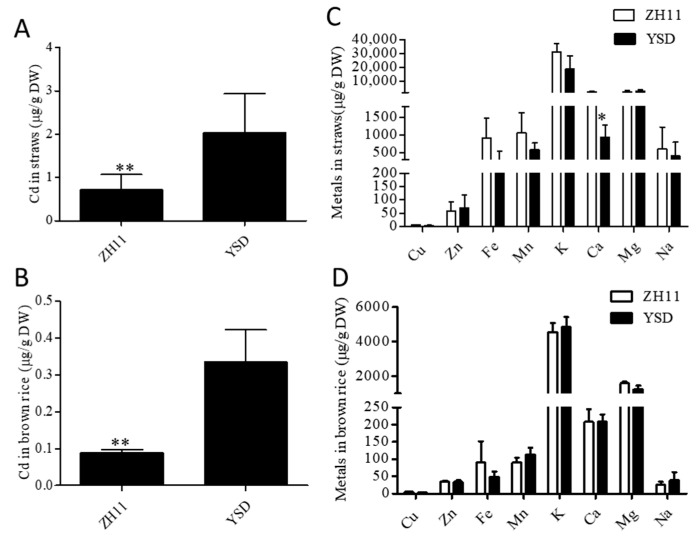
Cd content in YSD straw and grain was significantly higher than that in ZH11. Determination of Cd and other metal elements in YSD and ZH11 straw (**A**,**C**) and brown rice (**B**,**D**) at ripening stage grow in 0.4 mg/kg Cd-contaminated field. Data are presented as the means (±SD), n = 5. Significant differences were determined using the Student’s *t*-test: * *p* < 0.05, ** *p* < 0.01.

**Figure 2 plants-12-01226-f002:**
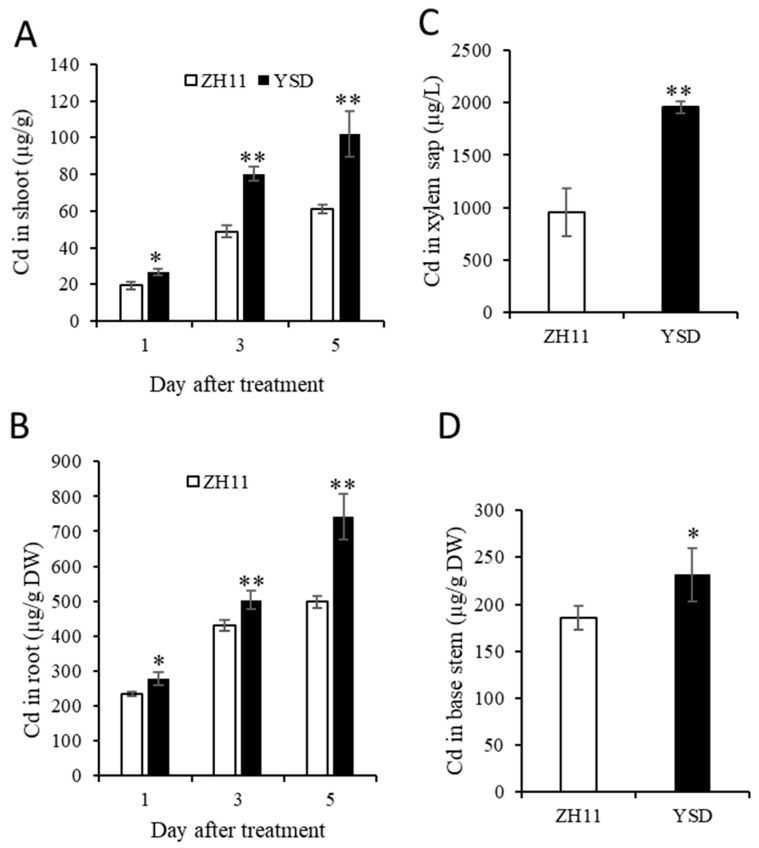
Time-dependent higher Cd accumulation in shoots and roots of YSD than ZH11. After 4 weeks of hydroponics, rice seedlings were treated with 10 μM Cd for 1, 3, and 5 days, Cd content in the shoot (**A**), root (**B**), basal stem (**C**) and xylem sap (**D**) were determined by ICP-MS. Data are presented as the means (±SD), n = 5. Significant differences were determined using the Student’s *t*-test: * *p* < 0.05, ** *p* < 0.01.

**Figure 3 plants-12-01226-f003:**
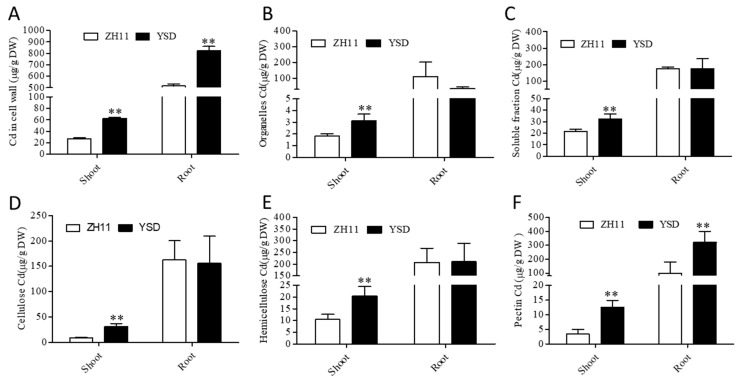
Cd content in different subcellular and cell wall components of YSD and ZH11. After 4 weeks of hydroponics, rice seedlings were treated with 10 μM Cd for 3 days, Cd content in cell wall (**A**), organelles (**B**), soluble part (**C**), cellulose (**D**), hemicellulose (**E**), and pectin (**F**) of the shoot and root were determined by ICP-MS. Data are presented as the means (±SD), n = 3. Significant differences were determined using the Student’s *t*-test: ** *p* < 0.01.

**Figure 4 plants-12-01226-f004:**
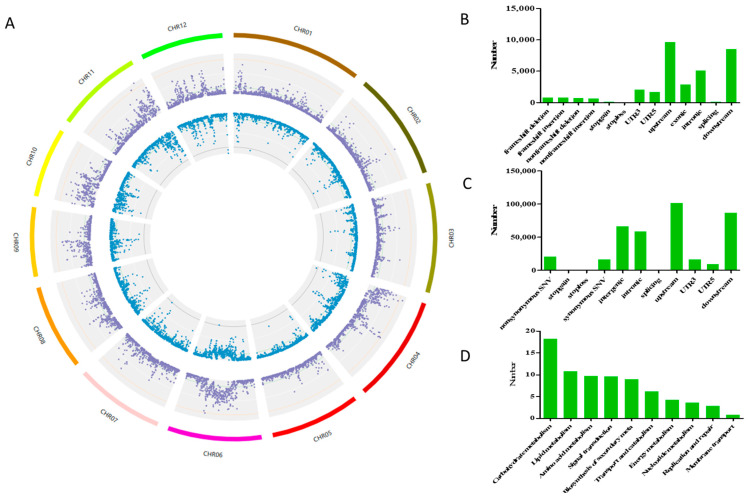
Genome-level variation of YSD compared to ZH11. Fresh leaves of 10-day-old plants were sampled for isolation of genomic DNA (gDNA) to perform whole-genome resequencing. (**A**) Genome variation Circos diagram. Circle 1: The chromosome; Circle 2: The purple locus represents the distribution of SNP density in the genome; Circle 3: Blue sites represent fractions of genomic InDel density. (**B**) Statistical map of SNP annotation results at gene level. (**C**) Map of InDel annotation results at gene level, and (**D**) pathway analysis of mutated gene loci.

**Figure 5 plants-12-01226-f005:**
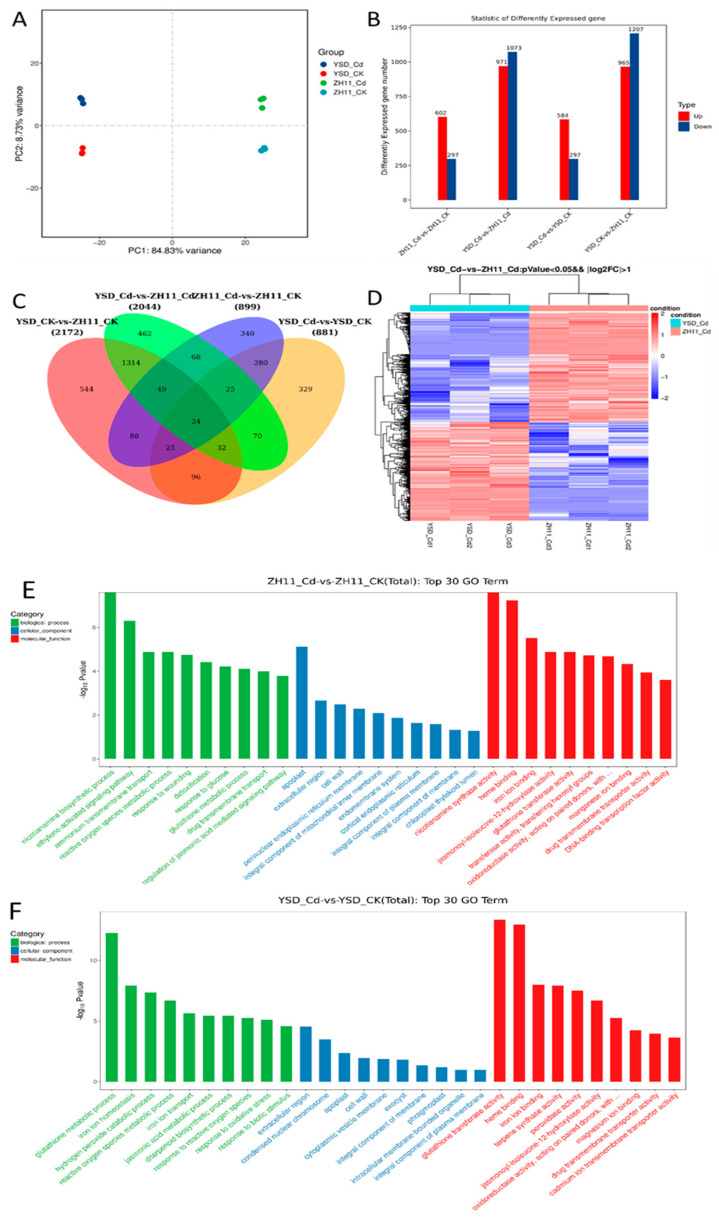
Analysis of transcriptome data from YSD and ZH11 roots. After 4 weeks of hydroponics, rice seedlings were treated with 0, 10 μM Cd for 3 days, and roots were taken for RNA_seq analysis. (**A**) Principal component analysis. (**B**) Number of up_ and down_regulated differentially expressed genes. (**C**) Venn diagram analysis of differentially expressed genes, (**D**) Heat map of differentially expressed genes. (**E**) GO annotation of differentially expressed genes in response to Cd stress in the roots of ZH11. (**F**) GO annotation of differentially expressed genes in response to Cd stress in the roots of YSD.

**Figure 6 plants-12-01226-f006:**
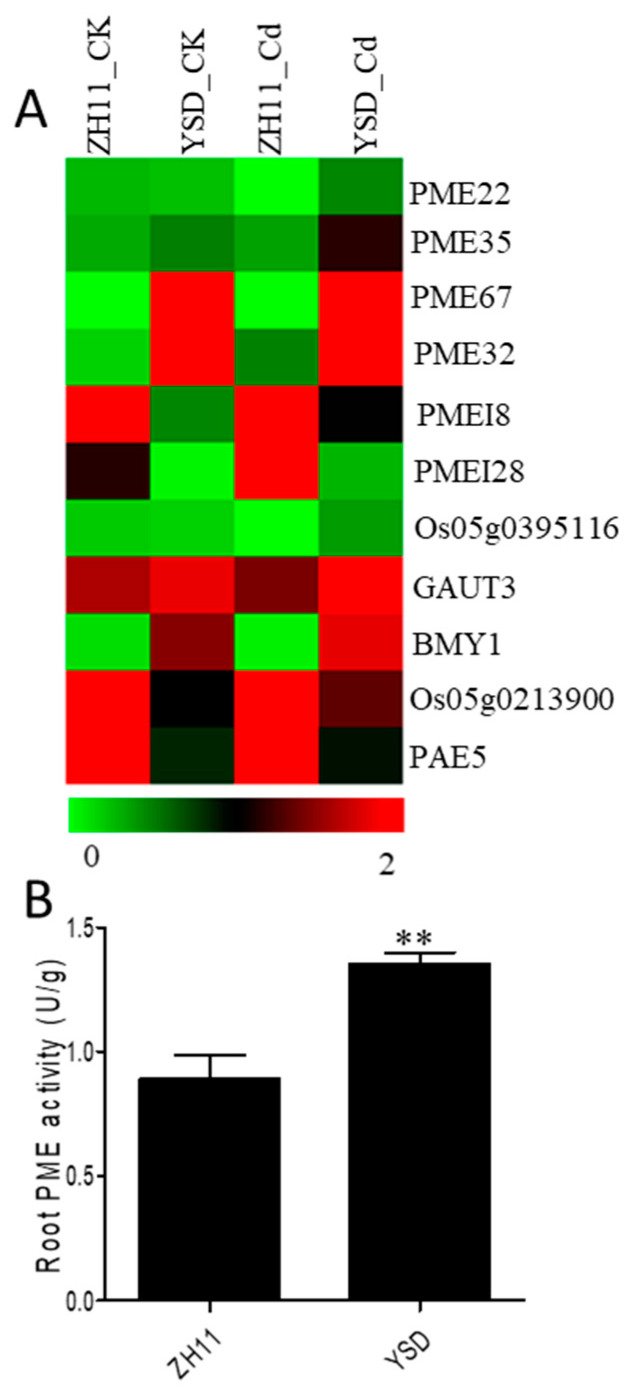
Differentially expressed genes (DEGs) involved in cell wall pectin metabolism and PME activity between YSD and ZH11 root under Cd treatments. (**A**) Transcriptional profiling of pectin methylesterase (PME 22/PME35/PME67/PME32), pectin methylesterase inhibitor protein (PMEI8/PMEI28), pectate lyase (Os05g0213900/PAE5) and pectin biosynthesis (CAUT3/BMY1). (**B**) PME activity in leaves of 30-day-old rice treated with 10 μM CdCl_2_ for 3 days. Data are presented as the means of 3 independent biological replicates (n = 3) and vertical bars represent the SD., ** indicates significant differences between the cultivars at *p* < 0.01.

**Figure 7 plants-12-01226-f007:**
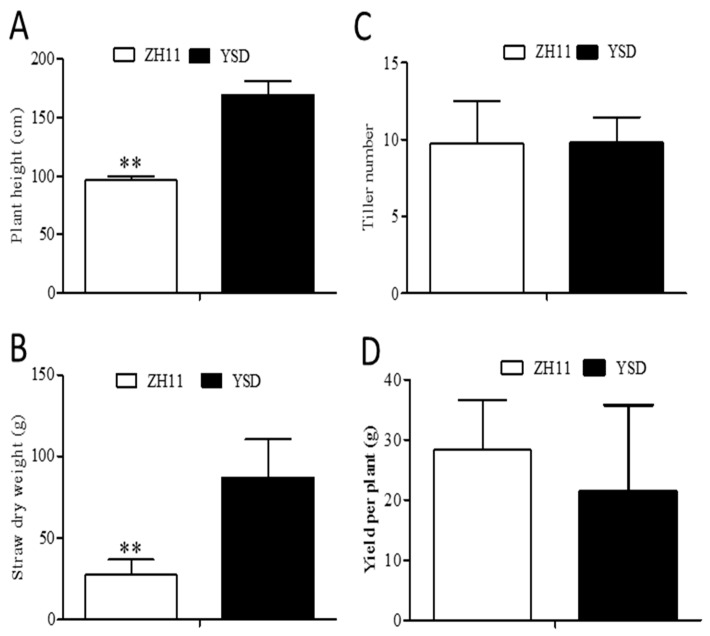
Analysis of agronomic characters of YSD and ZH11. Measurement of rice height (**A**), straw dry weight (**B**), tiller number (**C**) and yield per plant (**D**) at harvest stage. Data are presented as the means (±SD), n = 10. Significant differences were determined using the Student’s *t*-test: ** *p* < 0.01.

**Table 1 plants-12-01226-t001:** Cell-wall-associated variation in gene annotation.

GO Annotation	Number	Mutant Gene ID
Cell wall biogenesis	4	*Os03g0172000*; *Os03g0172200*; *Os03g0172700*; *Os03g0172100*
Cell wall modification	8	*Os02g0558200*; *Os01g0160100*; *Os01g0188400*; *Os01g0857400*; *Os03g0584224*; *Os01g0168600*; *Os01g0159800*; *Os04g0600800*
Plant-type cell wall organization	6	*Os02g0156600*; *Os01g0918400*; *Os01g0917900*; *Os02g0816200*; *Os02g0139300*; *Os04g0449000*
Cell wall macromolecule catabolic process	4	*Os01g0357800*; *Os04g0116200*; *Os04g0166000*; *Os01g0224000*

## Data Availability

The data presented in this study are available on request from the corresponding author.

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
