# Peer review of "Genome and Transcriptome Identification of a Rice Germplasm with High Cadmium Uptake and Translocation"

_plants, 2023, doi:10.3390/plants12061226_

Round 1

Reviewer 1 Report

General comments:

The manuscript presents a detailed analysis of a Cd accumulating rice variety compared to control variety with the objective to understand the regulatory mechanism of Cd accumulation in rice as possible targets for phytoremediation. Analysis includes genome sequence comparison, RNA seq, phenotyping and determination of glutathione and nicotinamide content. Results indicate that cell wall modification genes, on sequence and expression level, might be involved in differential Cd accumulation. The manuscript is very well written. Comprehensive analysis support the conclusion.

Detailed comments

Abstract:

L17 – please define ZH11

L19 – Better: depending on sampling time

L23 – I would advise to add: Transcriptome analysis in Cd treated plants showed that…..

Introduction:

Please add full  names (NRAMP, HMA, LCT1…)

The introduction should include some more information about current methods of Cd phytoremediation

The introduction, in general is very short. Authors should give some more detailed information about the processes mentioned, such as the effect of salicylic acid on Cd in cell wall, boron and CD content in cell wall, OsCD1 in Cd accumulation.

Results:

L174-177: the in vitro experiment should be described in materials and methods

Author Response

Reviewer 1

General comments:

The manuscript presents a detailed analysis of a Cd accumulating rice variety compared to control variety with the objective to understand the regulatory mechanism of Cd accumulation in rice as possible targets for phytoremediation. Analysis includes genome sequence comparison, RNA seq, phenotyping and determination of glutathione and nicotinamide content. Results indicate that cell wall modification genes, on sequence and expression level, might be involved in differential Cd accumulation. The manuscript is very well written. Comprehensive analysis support the conclusion.

Response:Thanks to the reviewers for their positive and useful suggestions

Detailed comments

Abstract:

L17 – please define ZH11

Response:Zhonghua 11(ZH11) is a commonly used japonica rice variety. Line 18.

L19 – Better: depending on sampling time

Response:Revised as suggested. Line 20

L23 – I would advise to add: Transcriptome analysis in Cd treated plants showed that…..

Response:Revised as suggested. Line 24-25

Introduction:

Please add full names (NRAMP, HMA, LCT1…)

Response:We had added the full names (NRAMP, HMA, LCT1…) in the revised manuscript.  Line 50-51,54,58,59-60.

The introduction should include some more information about current methods of Cd phytoremediation

Response:We had added some information about current methods of Cd phytoremediation in the introduction. Line 46-48.

The introduction, in general is very short. Authors should give some more detailed information about the processes mentioned, such as the effect of salicylic acid on Cd in cell wall, boron and CD content in cell wall, OsCD1 in Cd accumulation.

Response:Detailed information about the processes mentioned had been added in the revised introduction. Line 58-62, 69-74.

Results:

L174-177: the in vitro experiment should be described in materials and methods

Response:The in vitro experiment had been described in materials and methods section. Line 307-310.

Reviewer 2 Report

This article is devoted to the study of the characteristics of the wild type of rice in order to use it for phytoremediation and as a genetic resource for studying various mechanisms involved in the accumulation of cadmium. the use of rice for phytoremediation is relevant for China and Japan. 

The work was carried out at a high technical level. Object is interesting and promising. To improve the perception of the work, it is necessary to make some clarifications. Please, see the attached file.

Author Response

Reviewer 2

This article is devoted to the study of the characteristics of the wild type of rice in order to use it for phytoremediation and as a genetic resource for studying various mechanisms involved in the accumulation of cadmium. the use of rice for phytoremediation is relevant for China and Japan.

Response:Thank the reviewers for their positive comments.

The work was carried out at a high technical level. Object is interesting and promising. To improve the perception of the work, it is necessary to make some clarifications. Please, see the attached file.

Response: Thank the reviewers for their valuable suggestions.

  • if you are talking about the wild type YSD, what type ZH11 and how does it differ from the wild

Response: We redefined ZH11 and YSD rice varieties in the revised manuscript. Line 16,18.

  • it is necessary to justify why these rice varieties are taken. previously have data. to expand the justification of the purpose of the work

Response: We described the purpose and content of these works in detail, proposed the starting point of our research. Line 58-62, 69-76.

  • 10 μM CdCl2 Or 0.4mg/kg Cd

Response: This has been clarified in the revised manuscript. Line 88.

  • field experiments and laboratory divided for better understanding

Response: Field experiments and laboratory had been divided in the revised method. Line 289-295, 296-299.

  • Analysis of agronomic characters

Response: Analysis of agronomic characters had been added in the methods section. Line 289, 293-295.

  • Whole-genome resequencing reference

Response: Whole-genome resequencing reference had been added in methods section. Line 326, 436-438.

  • Determination of glutathione and nicotinamide content reference

Response: Determination of glutathione method, and nicotinamide content reference had been added in methods section. Line 340-341, 439-441.
